# The Radial Electric Field Excited Circular Disk Piezoceramic Acoustic Resonator and Its Properties

**DOI:** 10.3390/s21020608

**Published:** 2021-01-17

**Authors:** Andrey Teplykh, Boris Zaitsev, Alexander Semyonov, Irina Borodina

**Affiliations:** Kotel’nikov Institute of Radio Engineering and Electronics of RAS, Saratov Branch, 410019 Saratov, Russia; zai-boris@yandex.ru (B.Z.); alex-sheih@yandex.ru (A.S.); borodinaia@yandex.ru (I.B.)

**Keywords:** acoustic resonator, circular piezoceramic disk, finite element method, electrical impedance, acoustic resonance spectroscopy, electromechanical coupling coefficient, Q-factor, Nelder–Mead algorithm

## Abstract

A new type of piezoceramic acoustic resonator in the form of a circular disk with a radial exciting electric field is presented. The advantage of this type of resonator is the localization of the electrodes at one end of the disk, which leaves the second end free for the contact of the piezoelectric material with the surrounding medium. This makes it possible to use such a resonator as a sensor base for analyzing the properties of this medium. The problem of exciting such a resonator by an electric field of a given frequency is solved using a two-dimensional finite element method. The method for solving the inverse problem for determining the characteristics of a piezomaterial from the broadband frequency dependence of the electrical impedance of a single resonator is proposed. The acoustic and electric field inside the resonator is calculated, and it is shown that this location of electrodes makes it possible to excite radial, flexural, and thickness extensional modes of disk oscillations. The dependences of the frequencies of parallel and series resonances, the quality factor, and the electromechanical coupling coefficient on the size of the electrodes and the gap between them are calculated.

## 1. Introduction

There are many types of piezoelectric transducers and resonators that operate in different acoustic modes and are used in different practical applications. Cylindrical piezoelectric transducers operating on a longitudinal acoustic mode are used for the emission and receiving of ultrasound in a liquid [1,2] and piezoelectric energy harvesting solutions [3,4]. Resonators with a transverse electric field are used for various sensor applications [5,6,7,8,9]. Resonators of this type are sensitive to mechanical and electrical properties of the medium, which borders on the free surface of the resonator [7,8].

One of the possible new applications of piezoceramic resonators is the creation of sensors for determining the acoustic and electrical characteristics of the medium in contact with the resonator by the method of broadband acoustic resonance spectroscopy (ARS) [9]. This work is devoted to the study of a new type of piezoelectric resonator with ring electrodes, which can be used to determine the mechanical and electrical properties of thin films or viscous conducting liquids. The study of such a resonator was carried out in two stages. First, the electrical impedance of the unloaded resonator was measured in a sufficiently wide frequency range, and all its resonant frequencies were found. At this stage, a finite element model of the resonator was created, and the characteristics of the material (elastic, piezoelectric and dielectric constants, density, and viscosity) of the resonator itself were refined [10,11]. This was done by the method of broadband acoustic resonance spectroscopy [10,11,12]. Then, these measurements were carried out in the same frequency range for the resonator, the free surface of which was in contact with the investigated medium. This medium can be a viscoelastic film with finite conductivity or viscous conducting liquid. By changing the values of the resonance frequencies and the magnitude of the resonance peaks, one can estimate the properties of the medium under study [12]. In order to determine the mechanical and electrical properties of the medium, the electric field that accompanies acoustic oscillations must freely penetrate from the resonator into the medium. This means that the medium must be in direct contact with the resonator material and not with a metal electrode on its surface. To successfully apply the method of broadband acoustic resonance spectroscopy, it is necessary to be able to effectively determine the oscillation spectrum of the resonator (its natural frequencies) or the response of the resonator (its electrical impedance) to excitation at a certain frequency [13]. Our studies revealed that the distribution of the acoustic and electric fields and the electric impedance of a piezoelectric disk made of a 6 mm group piezomaterial, the crystallographic axis of which coincides with the axis of the disk, can be calculated quite accurately and quickly. In this case, it is possible mathematically to strictly take into account the different positions of the exciting electrodes and the inhomogeneity of the disk material if this does not violate the axial symmetry of the problem [14]. In this paper, we propose a design of such a resonator in the form of a circular piezoceramic disk with a radial exciting electric field. A mathematical model of this resonator based on the finite element method was created. A comparison of theoretical and experimental results for radial electric field excited resonator made from Russian commercially available piezoelectric ceramics with barium lead zirconate titanate (BPZT) was carried out, and the material constants of piezoelectric ceramics were refined. The dependences of its electrical impedance on frequency and other characteristics of the resonator were calculated for different radii of the electrodes and the gap between them.

## 2. Materials and Methods

### 2.1. Numerical Model of a Radial Electric Field Excited Piezoceramic Disk Resonator

Consider the problem of forced vibrations of a circular piezoceramic disk excited by a pair of concentric electrodes located on one side of the disk. The geometry of the problem is shown in Figure 1.

Let us consider a disk with a diameter *d* and a thickness *h* made of piezoelectric ceramics belonging to the crystallographic class of 6 mm. Let the ceramic polarization axis be parallel to the *z* axis of the disk. Concentric metallic electrodes 1 and 2 are located on the lower side of the disk. Electrode 1 has radius *e*_1_, and its center coincides with the center of the disk. Electrode 2 has the shape of a ring with an inner radius *e*_2_ and extends to the outer edge of the disk. The gap between the electrodes is *g*, therefore the identity *e*_1_ + *g* + *e*_2_ = *d*/2 holds true.

The side surface and the upper side of the disk are mechanically and electrically free. The upper side of the disk may be adjacent to the test medium. Due to the absence of metal surfaces on the upper side of the disk, the electric field freely penetrates the space above the disk.

The mathematical problem is to find the distribution of the acoustic and electric fields inside the disk [15]. Due to the fact that the axis of the disk is parallel to the axis of polarization of the piezoceramic, this problem is axisymmetric and can be written in two-dimensional form relative to the cylindrical coordinates *r* and *z*. Then the desired solution to the problem can be written as
(1)$$\begin{array}{r} u_{r}=u_{r}\left(r,z\right)\exp\left(I\omega t\right)\\ u_{z}=u_{z}\left(r,z\right)\exp\left(I\omega t\right)\\ \varphi =\varphi \left(r,z\right)\exp\left(I\omega t\right) \end{array}\},$$

Here, *u_r_*, *u_z_* is the radial and axial components of mechanical displacement, *φ* is the electrical potential, *I* is the imaginary unit, *ω* is the circular frequency, and *t* is the time. In this case, the solution does not depend on the coordinate *θ*; therefore, the tangential component of the mechanical displacement *u**_θ_* is everywhere identically equal to 0 and may be omitted.

Thus, in this problem, it is necessary to take into account only four components of the deformation *S*,
(2)$$\left\{S\right\}=\left\{\begin{array}{c} S_{rr}\\ S_{\theta \theta }\\ S_{zz}\\ 2S_{rz} \end{array}\right\}=\left[\begin{array}{cc} \frac{\partial }{\partial r} & 0\\ \frac{1}{r} & 0\\ 0 & \frac{\partial }{\partial z}\\ \frac{\partial }{\partial z} & \frac{\partial }{\partial r} \end{array}\right]\left\{\begin{array}{c} u_{r}\\ u_{z} \end{array}\right\}=\left[L_{u}\right]\left\{u\right\},$$
and two components of the electrical field *E*.
(3)$$\left\{E\right\}=\left\{\begin{array}{c} E_{r}\\ E_{z} \end{array}\right\}=-\left[\begin{array}{c} \frac{\partial }{\partial r}\\ \frac{\partial }{\partial z} \end{array}\right]\varphi =-\left[L_{\varphi }\right]\varphi$$

In the considered two-dimensional axisymmetric case, some rows and columns can be removed from the tensors of material constants, and the corresponding tensors can be written in the following matrix form [14]:(4)$$\left[c\right]=\left[\begin{array}{cccc} c_{11} & c_{12} & c_{13} & 0\\ c_{12} & c_{11} & c_{13} & 0\\ c_{13} & c_{13} & c_{33} & 0\\ 0 & 0 & 0 & c_{44} \end{array}\right]\left(1+I\omega \eta \right),\;\left[e\right]=\left[\begin{array}{cccc} 0 & 0 & 0 & e_{15}\\ e_{31} & e_{31} & e_{33} & 0 \end{array}\right],\;\left[\varepsilon \right]=\left[\begin{array}{cc} \varepsilon_{11} & 0\\ 0 & \varepsilon_{33} \end{array}\right].$$

Thus, the resonator material is fully characterized by five elastic constants *c*_11_, *c*_12_, *c*_13_, *c*_33_, *c*_44_, three piezoelectric constants *e*_15_, *e*_31_, *e*_33_, two dielectric permittivity constants *ε*_11_, *ε*_33_, density *ρ*, and scalar viscosity factor *η*. The problem is described by the system of the following equations [14]:(5)$$\begin{array}{r} -\omega^{2}\int_{V}\rho {\left\{u\right\}}^{T}\left\{u\right\}dV+\int_{V}{\left([L_{u}]\left\{u\right\}\right)}^{T}[c][L_{u}]\left\{u\right\}dV+\int_{V}{\left([L_{u}]\left\{u\right\}\right)}^{T}[e][L\varphi ]\varphi dV=0\\ \int_{V}{\left([L_{\varphi }]\varphi \right)}^{T}[e][L_{u}]\left\{u\right\}dV-\int_{V}{\left([L_{\varphi }]\varphi \right)}^{T}[\varepsilon ][L_{\varphi }]\varphi dV=0 \end{array}\}.$$

Here *L_u_*, *L_φ_* are the linear differential operators with respect to *u* and *φ*, respectively, and upper index *T* means transposition.

The following boundary conditions are applied. A mechanical boundary condition is specified on the disk axis:(6)$${u_{r}=0|}_{r=0}.$$

Assuming that an HF electric voltage of amplitude *V* is applied to the electrodes, the electrical boundary conditions on the electrodes can be written as
(7)$${\varphi =+V|}_{r\in e_{1}},\;{\varphi =-V|}_{r\in e_{2}}$$

The rest of the disk surface is mechanically and electrically free.

The solution of this problem by using the two-dimensional finite element method allows us to calculate the electrical impedance of the disk for a given frequency, taking into account the known or prescribed material constants of piezoceramics and the geometry of the resonator [4]. The solution of the inverse problem for this case will make it possible to perform the refinement of the material constants of the piezoceramic.

### 2.2. Creation of a Radial Electric Field Exciting Piezoceramic Resonator from Piezoelectric Ceramics of the BPZT Type

The blanks for the resonators as disks made from barium lead zirconate titanate (BPZT) piezoceramics together with the deposited silver electrodes were purchased from OOO Avrora-ELMA, Volgograd, Russia. The commercial name of this ceramics is VA-650, its chemical composition is Pb_0.75_Ba_0.25_(Zr_0.53_Ti_0.47_)O_3_. Due to the fact that the resonator’s own electrodes had a sufficiently large thickness (~20 microns), the old electrodes were removed from both sides of the resonator in two steps. In the first step, most of the metal was removed by a flat grinding abrasive wheel. In the next step, a thin layer of the remaining metal was removed using a cotton swab with 50% aqueous nitric acid solution.

After removing the electrodes, the geometric dimensions and mass of the resonator were measured. The resonator diameter was measured with a vernier caliper, and the thickness was measured with a micrometer probe (average of five measurements at different points). The mass was determined using the PA214C (OHAUS Corporation, NJ, USA) electronic balance. The measurement results are shown in Table 1.

The deposition of new aluminum electrodes was carried out by the method of vacuum sputtering on a vacuum universal post VUP-5 setup using a specially made mask in the form of a nickel ring. The ring was fixed on the top surface of the piezoceramic disk using a permanent magnet. The thickness of the obtained electrodes was about 2000 Å. The concentricity of the created electrodes was monitored under an optical microscope. The positioning error did not exceed 20 μm.

To create an electrical contact, gluing a thin gold wire with a diameter of about 20 microns was carried out using a special conductive glue of the Silver print brand (GC Electronics, USA). The wire connected to the central electrode *e1* was glued exactly to the center of the disk.

Before measurement, the resonator was fixed in a special holder, which is shown in Figure 2, and was held on a low impedance foam backing.

### 2.3. Measurement of Electrical Impedance of Radial Electric Field Excited Piezoceramic Resonator in Wide Frequency Range

The real and imaginary components of the electrical impedance of the radial field resonator were measured using an impedance meter E4990A (Keysight Technologies, Santa Rosa, CA, USA) after proper pre-calibration. The measurements were carried out in the frequency range 1–1001 kHz with a step of 10 Hz; therefore, a total of 100,001 points were measured. Since the E4990A impedance meter cannot measure more than 1601 points in one specified range, a special control program—Standard Commands for Programmable Instruments (SCPI)—was used to automatically switch frequency ranges. This made it possible to fully automate the measurement process and provide a frequency measurement step of 10 Hz. Measurement of all 100,001 reports in the range 1–1001 kHz took 330 s; each measurement cycle was repeated three times, and then the average values were calculated. During the entire measurement time, the piezoresonator was kept in a thermostat at a constant temperature of 25 °C ± 0.1 °C, and the relative air humidity did not exceed 20%. Measurement results are shown in Figure 3.

These measurements allowed estimating to estimate the viscosity factor of piezoelectric ceramics *η* from the shape of the peaks of the resonance curve [16]
(8)$$\eta =\frac{\Delta \omega }{\omega_{\max}^{2}},$$
where *ω*_max_ is the circular resonance frequency and ∆*ω* is the width of the resonance peak at level 0.707 of maximum. The values of *η* were calculated for all resonance peaks observed in the measuring range. The minimal value of *η* is shown in Table 1 and used in subsequent calculations.

The results of measurements of resonator characteristics for resonator under study are shown in Table 1.

### 2.4. Solution of the Inverse Problem and Refinement of the Material Constants of the Piezoceramic of the Resonator

Using the method described in Section 2.1, we simulated a radial electric field excited piezoceramic resonator. Its geometry, i.e., the diameter, thickness, and exact location of both electrodes exactly corresponded to the experimental sample and are given in Table 1. This made it possible to solve the so-called direct problem, i.e., find the distribution of the acoustic and electric fields inside the piezoresonator at a given frequency of the exciting field, taking into account the given material constants of the piezoceramics.

The model parameters were 10 material constants of piezoceramics (i.e., five independent elastic constants *c*, three independent piezoelectric constants *e*, and two independent values of dielectric permittivity *ε*), which will be denoted later as *p_j_*, where *j* is the index of constant, *j* = 1, …, 10. Other parameters, named material density *ρ*, viscosity factor *η*, and geometry of the piezoresonator were measured directly for the sample to be measured using the method described above. The calculation result is shown in Figure 4. The FEM index denotes the result of the calculation by the finite element method, and the EXP index refers to the experimentally measured values. As can be seen, these frequency dependences are qualitatively the same; however, the exact resonance frequencies *f*_FEM_ calculated noticeably differ from the experimentally measured frequencies *f*_EXP_.

This can be explained by the discrepancy between the material constants used in the calculation and the “real” material constants of this particular sample of piezoceramics. Therefore, it is necessary to solve the “inverse problem” and refine the material constants of the piezoelectric ceramics of a particular resonator using its experimentally measured frequency dependence of the electrical impedance Z_EXP_. In the present work, this was done in two stages.

At the first stage, the degree of influence of each of the 10 material constants on the frequencies of parallel and serial resonance for all acoustic modes of the piezoresonator was determined. For this, a number of preliminary calculations were performed. Since different material constants have numerical values of various magnitude from 1 to 10^10^, each material constant *p_j_* in the course of further calculations was represented by the dimensionless parameter *d_j_* according to the following law:(9)$$p_{j}(d_{j})=p_{0j}\left(1+d_{j}/100\right),$$
where *p_j_* is the actual value of corresponding material constant of index *j*, *p*_0*j*_ is the reference value of respective material constant, therefore all *d_j_* values have the same order of magnitude about 10.

The preparatory calculation took place as follows. For each index *j* = 1, …, 10 the material constant *p_j_* varied within 90%, …, 110% of the reference value *p*_0*j*_, i.e., the value of *d_j_* varied within the range −10 ≤ *d_j_* ≤ 10. At the same time, the remaining constants with indexes *k* ≠ *j* were equal to the reference value, i.e., *d_k_* = 0. This process, which is called sensitivity analysis, can be illustrated in Figure 4.

This allowed for the frequency of each parallel and series resonance *f^(i)^*, where *i* is the resonance index for determining the dependence *f^(i)^(d_j_)* for each constant *p_j_*. In the present case, *i* = 1, ..., 19. The resulting theoretical dependence of the frequency *f*_FEM_^(i)^ was determined by the finite element method on *d*, where *d* denotes the entire set of *d_j_*, *j* = 1, ..., 10, which was presented as follows:(10)$$f_{\mathrm{FEM}}^{\left(i\right)}(d)\approx f_{0\mathrm{FEM}}^{\left(i\right)}+\sum_{j=1}^{10}a_{j}^{\left(i\right)}d_{j}+b_{j}^{\left(i\right)}d_{j}^{2}+c_{j}^{\left(i\right)}d_{j}^{3},$$
where *f*_0FEM_*^(i)^* is the frequency with index *i* and calculated using the reference set of material constants, and the values of *a_j_^(i)^*, *b_j_^(i)^*, and c*_j_^(i)^* were determined using the least-squares method. Then the objective function was defined as follows:(11)$$F_{1}(d)=\sum_{i=1}^{n}{\left(f_{\mathrm{FEM}}^{\left(i\right)}(d)-f_{\mathrm{EXP}}^{\left(i\right)}\right)}^{2}.$$

This function *F*_1_ was then minimized with respect to *d* by the well-known Nelder–Mead algorithm [17] also known as the simplex method. The advantage of this algorithm is that it does not require calculating the derivatives of the function to be minimized, which reduces the required amount of computation. This operation made it possible to determine the optimal set of *d_j_*, which corresponds to the minimum deviation of the theoretical values of frequencies from the experimental ones. In this case, the difference in the values of electrical impedance *R*, electrical admittance *G,* or absolute value of electrical impedance *Z* at resonant frequencies was not taken into account. As it turned out, this gives a very good initial approximation for the second stage of the refinement procedure.

At the second stage, the values of the material constants, which were determined at the previous stage of the work, were further refined. At this stage, the objective function, which was minimized by the algorithm, was written in the following form [18,19]:(12)$$F_{2}(d)=\sqrt{\sum_{i=1}^{n}{\left(\log\left|Z_{\mathrm{FEM}}^{\left(i\right)}/Z_{0}\right|-\log\left|Z_{\mathrm{EXP}}^{\left(i\right)}/Z_{0}\right|\right)}^{2}},$$
where |*Z*_EXP_*^(i)^*| и |*Z*_FEM_*^(i)^*| are the measured and calculated absolute values of the electrical impedance of piezoresonator excited on frequency *f_i_*, respectively, *Z*_0_ is the unit of electrical resistance equal 1 ohm (this is necessary to bring the dimension of the expression to one), and *n* is the number of points of measure. This function is well suited for finding the minimum because the expression under the sum sign has the same order of magnitude at all frequencies. As a result of the algorithm, the refined values of the material constants were obtained for each sample under study for the selected number of points *n* = 1000. The initial values for the algorithm (the initial simplex) were chosen randomly around the values determined at the previous stage with a relative deviation of no more than 1%. It is important to note that the use of a large initial deviation, as well as skipping the preliminary stage, often greatly increased the required number of iterations (sometimes by a factor of 10) to achieve convergence or led to finding another local minimum with clearly anomalous values of material constants.

The refined material constants of the resonator made of piezoelectric ceramics BPZT are given in Table 2.

The refined values of the material constants from the column named “Result of Stage 2” will be used in this work for further calculations.

## 3. Results

### 3.1. Comparison of Theoretical and Experimental Data for a Radial Electric Field Excited Piezoresonator Made of BPZT

After refining the values of material constants of the piezoelectric ceramics, the theoretical and experimental frequency dependences of the electric impedance of the resonator with the radial exciting field were compared. The comparison result is shown in Figure 5.

Figure 5 shows a lot of resonance peaks, which correspond to different modes of acoustic vibrations of the resonator. All acoustic modes corresponding to these peaks were successfully identified and labeled. It was found that this type of acoustic resonator supports three different types of acoustic modes.

The first type is radial modes *Rn*, which are characterized by a predominant displacement in the radial direction. The second type is thickness extensional modes *TEn*, which are characterized by a significant change in disk thickness. The third type is flexural modes *Fn*, which are characterized by a predominant displacement in the axial direction with a slight change in the disk thickness. As stated in [21], this mode can be excited only when the electrodes are disposed of so that the electric field generates the corresponding deformation. Therefore, for the same resonator but with a longitudinal exciting field, these modes will be absent. The index *n* denotes the mode number. The thickness extensional and flexural modes have a significant axial displacement component and therefore we would expect them to be suitable for emitting sound into a liquid [22]. The first two modes of each type are shown in Figure 6.

It should be noted that the radial and thickness extensional modes are typical for a disk resonator with a longitudinal exciting field [18,19]; however, flexural modes are not observed in such type of resonator.

A comparison was also made of the frequencies of parallel and series resonances, *Q*-factors, and electromechanical coupling coefficients [23]. The quality factor *Q* was calculated for parallel resonance using the following well-known formula [24]:(13)$$Q=\frac{f_{par}}{f_{+}-f_{-}},$$

Here, *f_par_* is the frequency of parallel resonance, *f*_+_ и *f*_−_ is the right and left frequencies of half-power, respectively.

In addition, for each vibration mode, the coefficient of electromechanical coupling *k*^2^ was calculated as follows [24]:(14)$$k^{2}=\frac{f_{par}^{2}-f_{ser}^{2}}{f_{par}^{2}},$$

Here, *f_par_* is the frequency of parallel resonance, *f_ser_* is the frequency of the series resonance for the same oscillation mode.

The characteristics of 19 resonant modes existing in this resonator in the range of 1–1001 kHz are presented in Table 3.

As can be seen from the presented data, as a result of the refinement of the material constants, the difference between the measured and calculated resonance frequencies does not exceed 4 kHz. The maximum *Q*-factor according to the experimental data exceeds 2400, and the electromechanical coupling coefficient reaches 6.69%, which is comparable with the results [18,19].

### 3.2. Study of the Dependence of the Resonator Characteristics on the Position and Width of the Gap between the Electrodes

Using the refined values of the material constants of the piezoceramics of the radial electric field excited piezoresonator, we studied the dependences of its resonance frequencies on the position of electrodes and width of the gap between them. For this, a resonator with a diameter *d* = 22 mm and a thickness *h* = 2 mm was modeled. The position of the electrodes on the bottom end of this resonator was changed in the following way.

Initially, the radius of the inner electrode *e*_1_ varied in the range 0.5–9.5 mm, the gap between the electrodes *g* was fixed and amounted to 1 mm. The dependences of the characteristics of the cavity for some modes for this case are shown in Figure 7. As can be seen from the presented data, the parallel resonance frequency changes several times more than the serial resonance frequency. The changes are non-monotonic and can contain several local maxima at a certain radius of the inner electrode.

Then, the gap *g* between the electrodes was changed in the range of 1–10 mm, the radius of the inner electrode *e*_1_ was 11-*g* mm. The dependences of the characteristics of the resonator for some modes for this case are shown in Figure 8. As can be seen from the presented data, in this case, the frequencies of both parallel and series resonances increase monotonically with an increase in the gap between the electrodes.

Gaps (absence of colored lines) in some plots at certain values of *e*_1_ and *g* mean that at a given position of the electrodes the corresponding mode is absent or its quality factor is *Q* < 1.

The flexural mode F0 mode has the maximum quality factor, its Q-factor can reach 12,000 (at a given value of the viscosity factor *η*). The quality factor of the rest of the modes ranges from several hundred to several thousand.

There is a change in the value of the electromechanical coupling coefficient determined by Equation (14) over a wide range. The maximum value of *k*^2^ is equal to 12.4% and is observed for the F5 mode (*f*_par_ = 544–590 kHz) at a gap value *g* = 5.8 mm.

## 4. Discussion

A new type of disk piezoceramic resonator was presented in the work—a resonator with a radial exciting field. The difference between the presented type of resonator and a disk resonator with electrodes at the ends of the disk (resonator with a longitudinal field) is that both electrodes are located at the same end of the disk. This leaves the second end of the disk free for contact with the medium under study, which makes such a resonator suitable for creating a sensor for both mechanical and electrical properties of the medium. In this case, the selected shape of the electrodes retains the axial symmetry of the disk, which greatly simplifies its mathematical modeling by a two-dimensional finite element method. This is the advantage of the presented type of resonators relative to resonators with a transverse exciting field with flat electrodes [5,6,7,8], which cannot be correctly modeled by a two-dimensional finite element method in a wide frequency range. The combination of these properties makes it possible to apply the methods of broadband ultrasonic spectroscopy both to the resonator itself to refine its characteristics and to the resonator in acoustic contact with the medium under study, which makes it possible to determine the acoustic and electrical characteristics of the medium using just one broadband measurement. The use of such a resonator to determine the properties (speed of sound, viscosity, dielectric permittivity, and electrical conductivity) of a viscoelastic film applied to the free end of the resonator, as well as the properties of a liquid, will be the subject of further research. The peculiarities of the investigated resonator should include the presence of flexural modes, which are absent in disk resonators with a longitudinal electric field [21]. Difficulties in working with resonators of this type lie in the need to ensure a constant temperature during the entire broadband measurement, which can be quite long.

## 5. Conclusions

This paper presents a new type of piezoceramic acoustic disk resonator with a radial exciting electric field. The advantage of this type of resonator is the localization of the electrodes at one side of the disk, which leaves the second side being free for the contact of the piezoelectric material with the surrounding medium. This allows using such a resonator as a sensor base for analyzing the properties of this medium. The problem of exciting such a resonator by an electric field of a given frequency is solved by using a two-dimensional finite element method. A method for solving the inverse problem for determining the characteristics of a piezomaterial from the broadband frequency dependence of the electrical impedance of a single resonator was shown. A comparison of experimental and theoretical results for a circular disk resonator with a diameter of 22 mm, a thickness of 2 mm, and made of Russian-made BPZT piezoelectric ceramics, was carried out. It is shown that the proposed finite element model well describes the characteristics of a real resonator in the range of 1–1001 kHz and allows us to find the refined material constants of piezoelectric ceramics. The acoustic field inside the resonator is calculated, and it is shown that such location of electrodes makes it possible to excite radial, flexural, and thickness extensional modes of the disk oscillations. The dependences of the frequencies of parallel and series resonances, the quality factor, and the electromechanical coupling coefficient on the sizes of the electrodes and the gap between them are calculated. It is shown that the electromechanical coupling coefficient and the Q factor for some modes with such excitation method are not worse than for a conventional disk resonator with a longitudinal electric field.

## Figures and Tables

**Figure 1 sensors-21-00608-f001:**
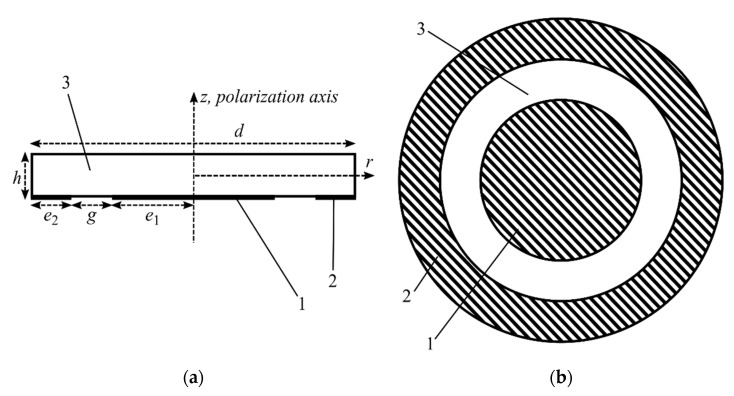
The geometry of the problem. (**a**) Side view of the resonator and (**b**) bottom view of the resonator; 1—internal electrode; 2—external electrode; 3—piezoceramic.

**Figure 2 sensors-21-00608-f002:**
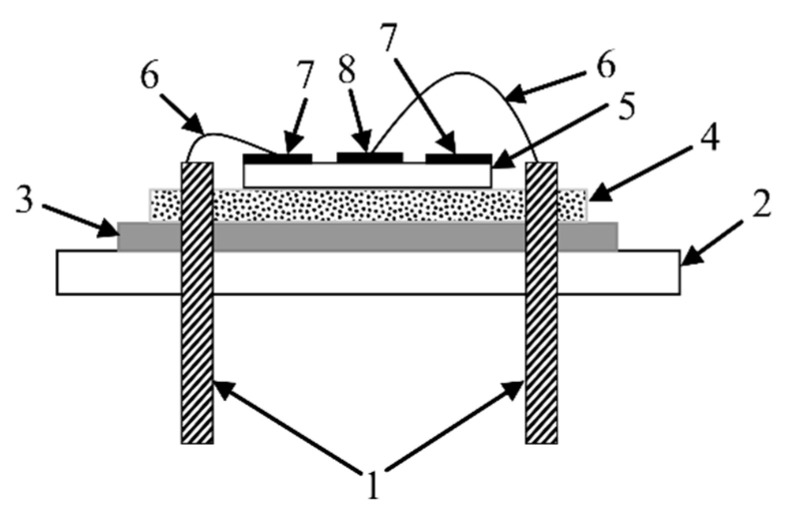
Holder for the resonator (side view): 1—copper contacts for connection to the measuring device (material; copper plate 1.2 mm); 2—plastic plate 3 mm thick; 3—cardboard 1 mm thick; 4—foam rubber 6 mm thick; 5—piezoceramic plate; 6—gold wire; 7—outer ring electrode of the resonator; and 8—inner round electrode of the resonator.

**Figure 3 sensors-21-00608-f003:**
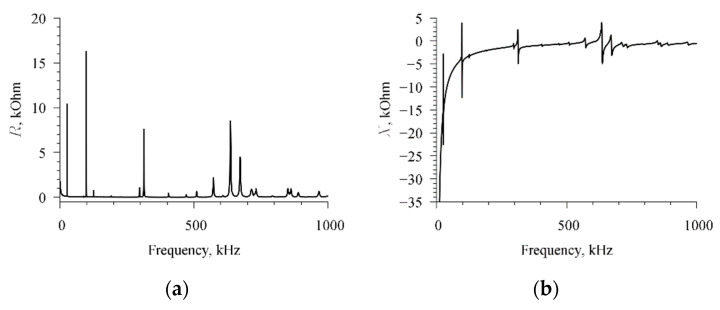
Results of measuring the frequency dependence of the electrical impedance of the resonator. (**a**) The real part of the electrical impedance and (**b**) the imaginary part of the electrical impedance.

**Figure 4 sensors-21-00608-f004:**
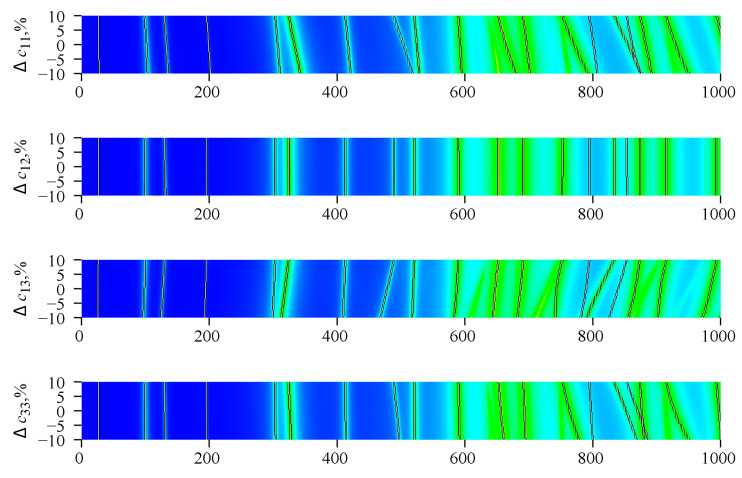
Sensitivity analysis for the elastic, piezoelectric, and dielectric material constants. Color maps correspond to the value of the real part of the electrical impedance *R*. The black lines with different slopes correspond to the local maximums of *R*.

**Figure 5 sensors-21-00608-f005:**
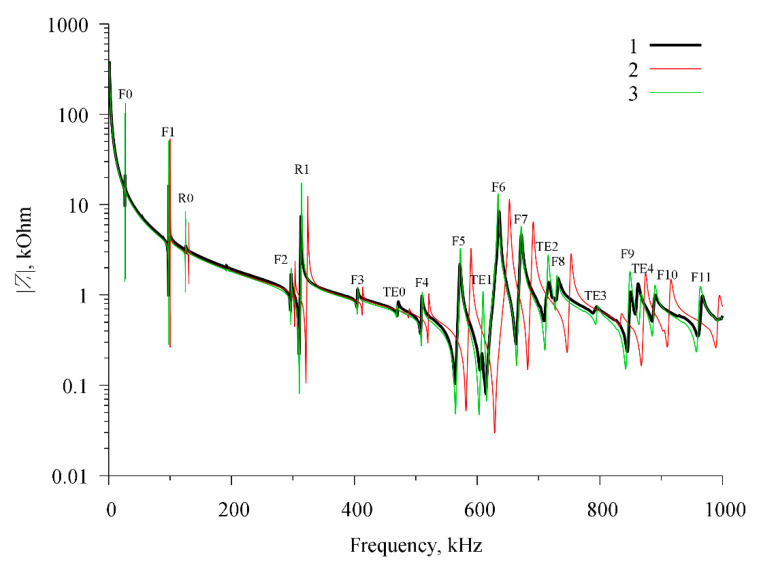
Frequency dependence of the modulus of electrical impedance of a radial electric field excited resonator. 1—experimental data; 2—calculation using the reference values of material constants; and 3—calculation using the refined material constants (stage 2).

**Figure 6 sensors-21-00608-f006:**
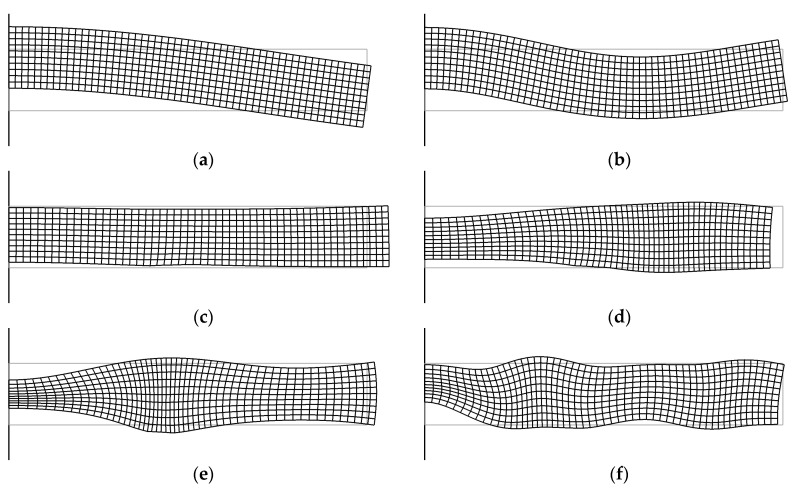
Distribution of mechanical displacements inside the disk resonator corresponding to different types of oscillations at different excitation frequencies: (**a**) flexural mode F0, *f* = 26,117 Hz; (**b**) flexural mode F1, *f* = 97,574 Hz; (**c**) radial mode R0, *f* = 124,871 Hz; (**d**) radial mode R1, *f* = 314,055 Hz; and (**e**) thickness extensional mode TE0, *f* = 469,336 Hz; and (**f**) thickness extensional mode TE1, *f* = 609,629 Hz. The vertical line to the left indicates the axis of the disk.

**Figure 7 sensors-21-00608-f007:**
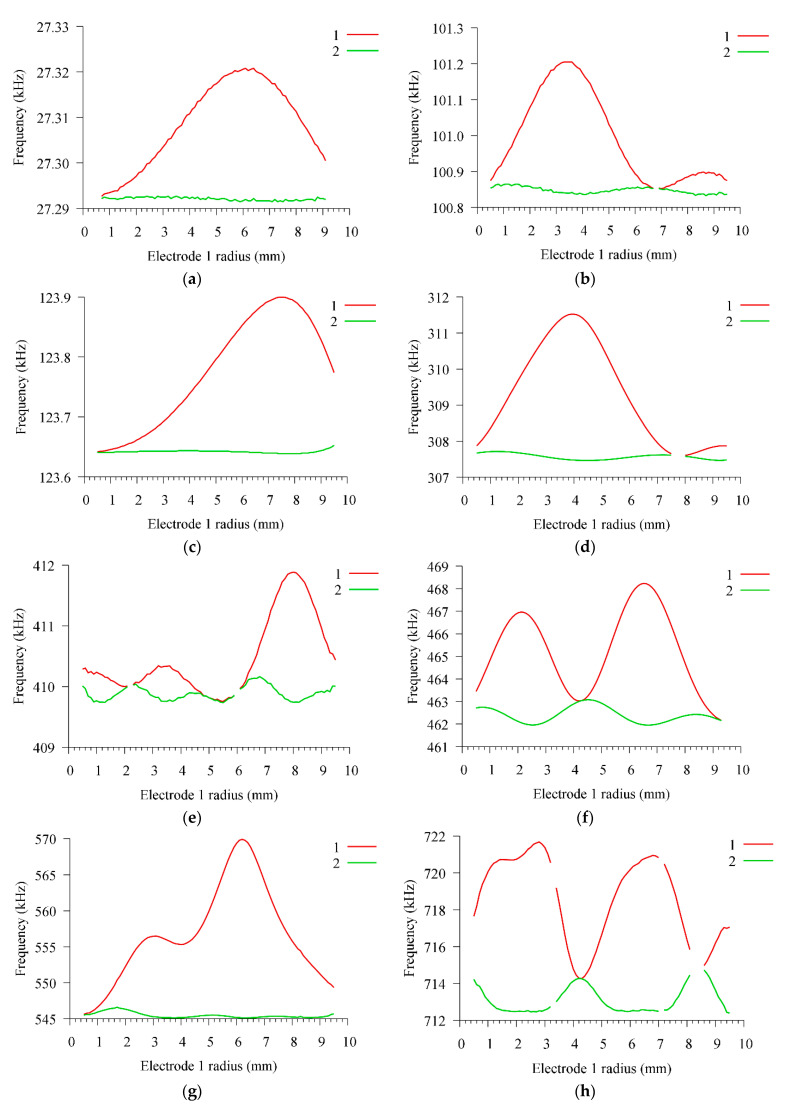
Dependences of the frequencies of (1) parallel and (2) series resonances on the radius of the inner electrode *e*_1_ for modes (**a**)F0, (**b**) F1, (**c**) R0, (**d**) R1, (**e**) F3, (**f**) TE0, (**g**) F5, and (**h**) TE2.

**Figure 8 sensors-21-00608-f008:**
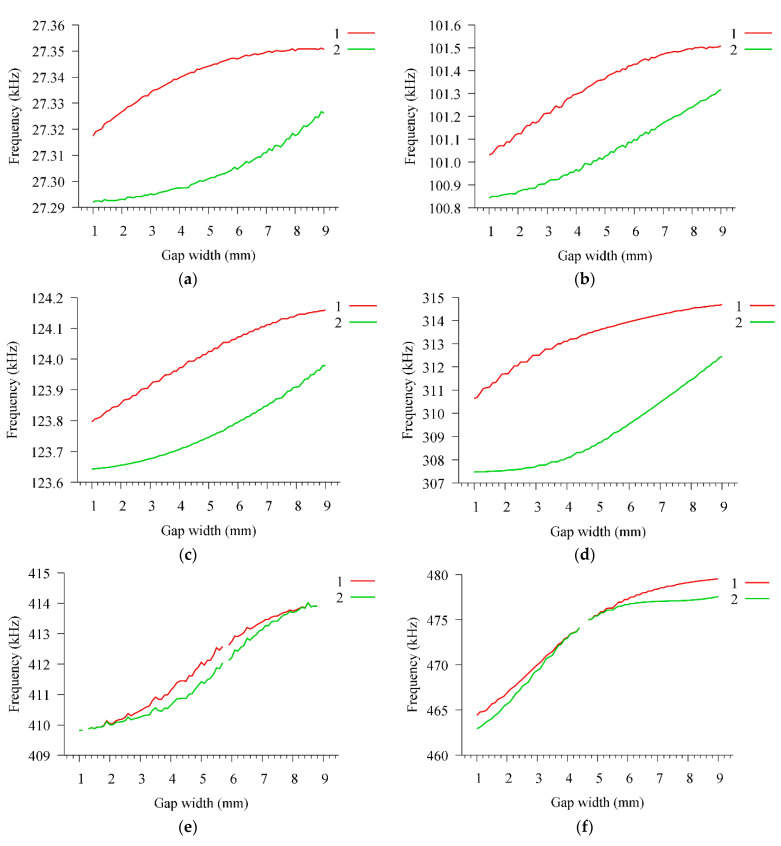
Dependences of the frequencies of (1) parallel and (2) series resonances on the width of the gap between the electrodes *g* for the modes (**a**) F0, (**b**) F1, (**c**) R0, (**d**) R1, (**e**) F3, (**f**) TE0, (**g**) F5, and (**h**) TE2.

**Table 1 sensors-21-00608-t001:** Measured characteristics of the resonator.

Characteristics Name	Value
Diameter (mm)	21.89
Thickness (μm)	1880
Mass (g)	5.1983
Density (kg/m^3^)	7347
Radius of inner electrode *e_1_*	4.06
Radius of outer electrode *e_2_*	5.04
*η*, s/rad	1 × 10^−9^

**Table 2 sensors-21-00608-t002:** The reference and refined material constants of the resonator made of piezoelectric ceramics barium lead zirconate titanate (BPZT).

Constant	Reference Value [20]	Result of Stage 1	Result of Stage 2
*c*_11_ × 10^10^ Pa	15.1	14.53	15.15
*c*_12_ × 10^10^ Pa	7.9	7.30	7.34
*c*_13_ × 10^10^ Pa	8.0	8.46	8.90
*c*_33_ × 10^10^ Pa	13.6	14.49	14.37
*c*_44_ × 10^10^ Pa	2.9	2.65	2.76
*e*_15_, C/m^2^	15.4	15.91	15.43
*e*_31_, C/m^2^	−7.9	−9.77	−10.17
*e*_33_, C/m^2^	17.7	14.58	14.55
ε_11_/ε_0_	1610	1610	1408
ε_33_/ε_0_	1280	1280	1198

**Table 3 sensors-21-00608-t003:** The mode name and parameters of 19 observed modes of acoustical oscillations in radial electric field excited piezoelectric resonator under study.

Mode	Fpar, kHz		Fser, kHz		Q		K^2^, %	
	EXP	FEM	EXP	FEM	EXP	FEM	EXP	FEM
F0	26.210	26.117	26.180	26.102	2440	4288	0.23	0.12
F1	97.740	97.574	97.310	97.266	702	3315	0.88	0.63
R0	125.560	124.871	125.530	124.781	719	2694	0.05	0.14
F2	296.660	297.051	296.220	296.730	521	1117	0.30	0.22
R1	313.010	314.055	310.160	310.320	503	957	1.81	2.36
F3	404.870	405.242	404.550	404.980	452	834	0.16	0.13
TE0	470.860	469.336	470.500	469.145	366	586	0.15	0.08
F4	509.590	510.406	508.500	509.512	378	656	0.43	0.35
F5	571.990	572.730	564.450	564.906	288	485	2.62	2.71
TE1	607.360	609.629	604.900	603.234	182	462	0.8	2.09
F6	635.760	634.246	614.110	615.582	313	499	6.69	5.80
F7	671.770	671.750	663.670	664.590	291	437	2.40	2.12
TE2	715.310	715.477	710.240	710.852	125	385	1.41	1.29
F8	731.340	729.387	730.100	728.039	311	408	0.34	0.37
TE3	793.100	795.309	790.750	794.406	148	343	0.59	0.23
F9	849.740	848.715	845.600	842.609	297	356	0.97	1.43
TE4	861.280	866.535	858.770	864.637	264	316	0.58	0.44
F10	888.590	889.074	886.690	886.305	252	352	0.43	0.62
F11	965.880	962.926	962.110	958.297	262	306	0.78	0.96

## Data Availability

Not applicable.

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
