# Peer review of "The Radial Electric Field Excited Circular Disk Piezoceramic Acoustic Resonator and Its Properties"

_sensors, 2021, doi:10.3390/s21020608_

Round 1

Reviewer 1 Report

This work presents a new type of piezoceramic acoustic disk resonator with a radial exciting electric field. The advantage of this type of resonators is the localization of the electrodes at one side of the disk, which leaves the second side being free for the contact of the piezoelectric material with the surrounding medium. This allows to use such a resonator as a sensor base for analyzing the properties of this medium. This work is interesting. However, some issues need to be addressed before considering acceptance.

  1. The device configuration looks like Quartz Crystal Microbalance (QCM). How about the moist influence on the output performance?
  2. How about the temperature impact on the output performance?
  3. As for Fig. 7 and Fig. 8, the unit, such as kHz and mm, should be expressed as (kHz) and (mm). No comma please.
  4. How about the long-term stability of the device?
  5. How did the structure and dimension of internal and external electrode affect the output performance of the device?
  6. Following the last question, is it possible to utilize COMSOL to simulate the electrode geometric effect on device performance?
  7. Some related article may enrich the background and concept in the introduction as references: ACS Nano, 2020, 14(5), 6067-6075., Nano Energy, 2020, 74, 104941. Adv. Funct. Mater. 2018, 28, 1704112ï¼›Joule 2017, 1, 480.

Reviewer 2 Report

Novel piezoceramic resonator was proposed to leave the space free for the piezoelectric material with a surrounding medium so this special configuration is useful to be characterized. The mathematical analysis and measured data with impedance graph is well matched. The K factor is reasonably achieved. The Q factor of the resonator is pretty high enough to be use.The electromechanical coupling coefficient is 6.69%.The frequency dependence is pretty stable. As mentioned, another end of the resonator could be useful to create the electric and mechanical properties as a sensor. The literature background search and simulation data for piezoceramic resonator could support the proposed new idea. There are no English grammar issues. Figure and Table quality are good to be clearly shown. However, authors miss many references so they need to be added. Therefore, the manuscript can be minor revision with the following minor comments.

1. Authors need to add the reference (At this stage, a finite element model of the resonator will be created and the characteristics of the material (elastic, piezoelectric and dielectric constants, density,viscosity) of the resonator itself could be refined).
2. Authors need to add the reference (To successfully apply the method of broadband acoustic resonance spectroscopy, it is necessary to be able to effectively determine the oscillation spectrum
of the resonator (its natural frequencies) or the response of the resonator (its electrical impedance) to excitation at a certain frequency.
3. Authors need to add the reference (There are many types of piezoelectric transducers and resonators that operate in different acoustic modes and used in different practical applications) with the reference (You, K., & Choi, H. ,Wide Bandwidth Class-S Power Amplifiers for Ultrasonic Devices. Sensors, 20(1), 290, 2020).
4. Authors need to add the reference (The mathematical problem is to find the distribution of the acoustic and electric fields inside the disk).
5. Authors need to add the reference (The solution of this problem by using the two dimensional finite element method allows us to calculate the electrical impedance of the disk, for a given frequency, taking into account the known or prescribed material constants of piezoceramics and the geometry of the resonator).
6.Authors need to add the reference (A comparison was also made of the frequencies of parallel and series resonances,Q-factors and electromechanical coupling coefficients) with the reference (Kim, J., You, K., Choe, S. H., & Choi, H. ,Wireless Ultrasound Surgical System with Enhanced Power and Amplitude Performances. Sensors, 20(15), 4165.2020).
7. Authors need to add the reference (The thickness extensional and flexural modes have a significant axial displacement component and therefore we would expect them to be suitable for emitting sound into a liquid).
